# Foliar Selenium Application to Reduce the Induced-Drought Stress Effects in Coffee Seedlings: Induced Priming or Alleviation Effect?

**DOI:** 10.3390/plants12173026

**Published:** 2023-08-23

**Authors:** Gustavo Ferreira de Sousa, Maila Adriely Silva, Mariana Rocha de Carvalho, Everton Geraldo de Morais, Pedro Antônio Namorato Benevenute, Gustavo Avelar Zorgdrager Van Opbergen, Guilherme Gerrit Avelar Zorgdrager Van Opbergen, Luiz Roberto Guimarães Guilherme

**Affiliations:** 1Soil Science Department, Federal University of Lavras, Lavras 37200-000, MG, Brazil; gustavoferreira_s@hotmail.com (G.F.d.S.); m.adriely@hotmail.com (M.A.S.); evertonmoraislp@gmail.com (E.G.d.M.); benevenutepedro@gmail.com (P.A.N.B.); gustavo.opbergen1@estudante.ufla.br (G.A.Z.V.O.); gerritavelar@gmail.com (G.G.A.Z.V.O.); 2Department of Biology, Federal University of Lavras, Lavras 37200-000, MG, Brazil; marianauesc@hotmail.com

**Keywords:** beneficial elements, oxidative stress, tropical agriculture, coffee belt, osmotic potential

## Abstract

This study aimed to investigate the role of Se supply in improving osmotic stress tolerance in coffee seedlings while also evaluating the best timing for Se application. Five times of Se foliar application were assessed during induced osmotic stress with PEG-6000 using the day of imposing stress as a default, plus two control treatments: with osmotic stress and without Se, and without osmotic stress and Se. Results demonstrated that osmotic stress (OS) promoted mild stress in the coffee plants (ψw from −1.5MPa to −2.5 MPa). Control plants under stress showed seven and five times lower activity of the enzymes GR and SOD compared with the non-stressed ones, and OS was found to further induce starch degradation, which was potentialized by the Se foliar supply. The seedlings that received foliar Se application 8 days before the stress exhibited higher CAT, APX, and SOD than the absolute control (−OS-Se)—771.1%, 356.3%, and 266.5% higher, respectively. In conclusion, previous Se foliar spray is more effective than the Se supply after OS to overcome the adverse condition. On the other hand, the post-stress application seems to impose extra stress on the plants, leading them to reduce their water potential.

## 1. Introduction

Atmospheric carbon dioxide (CO_2_) has increased over the past seven decades. It is correlated with gradual and systematic modifications in average climate conditions, such as temperature and precipitation variance [1]. Indeed, such extreme events (e.g., heat waves, floods, and severe drought seasons) expose the remarkable vulnerability of agricultural systems [2,3].

These environmental changes have modified temperature and rain patterns worldwide, making coffee cultivation uncertain in commonly cultivated areas [4,5]. Coffee is a crop sensitive to precipitation variability, and rainfall instability can lead to high losses in coffee production. Arabica coffee requires between 1000 and 2700 mm of annual precipitation and from one to three months of dry season annually [6]. Due to its temperature and humidity demands, coffee cultivation is limited to the intertropical region, commonly called the coffee belt [7]. 

The plant side effects of the lack of water in the crop system include drought stress [8]. Drought stress imposes osmotic stress (OS) due to the lack of water in the plant tissue. OS promotes changes in plants’ physiological, morphological, ecological, biochemical, and molecular traits [9,10]. Water deficit directly affects crops’ growth, development, and yield [11]. As an immediate response to OS, the stomata closes, which constrains the transpiration flow and the CO_2_ fixation. These responses vigorously reduce the photosynthetic rates and hence the production of photoassimilates [12]. The impact of OS on coffee plants reflects negatively in the harvest in progress and future ones [13].

Plant mineral nutrition is considered a strategy to reduce the adverse effects of OS. Selenium (Se) is one of the promising approaches to fight the metabolic responses in plants under this type of adverse condition [14,15,16]. Selenium is not a plant nutrient, but several studies have reported its beneficial effects, mainly under stress conditions (e.g., salinity, chilling stress, metals accumulation, and drought stress) [11,17,18,19]. The extensive antioxidant capacity of Se arises from its ability to enhance selenoproteins, like glutathione peroxidase. These selenoproteins play a crucial role in counteracting reactive oxygen species (ROS) generated during plant osmotic imbalance in challenging conditions. Thus, using selenium as an osmoprotective strategy may effectively alleviate the detrimental impact of abiotic stresses [20,21]. 

In a recent study, Rady et al. [22] found that Se was responsible for mitigating adverse effects of water-deficit stress conditions in *Solanum lycopersicum.* According to these authors, the improvements observed in response to Se indicate that it plays a regulatory role in plants under stress by positively influencing both enzymatic and non-enzymatic components of the plant’s antioxidative defense system. Selenium can also foster synergistic interactions with other nutrients, all of which contribute to improved resilience against abiotic stresses and overall better plant growth [23,24]. However, the study of the effect of isolated Se applications remains to be clarified under specific conditions as a first step to define its effects on the main crops of interest. 

As a result of Se application in plants, some authors have noticed an increase in shoot and root biomass and better plant development [25], as well as improved regulation in the status of water, and higher antioxidant apparatus activation in water-stressed crops [26]. Sousa et al. [19] found that Se can modulate nutrient uptake, carbohydrate breakdown, and enzymatic activity in coffee plants after low-temperature stress, helping the plants to overcome adverse conditions. Assessing the effect of foliar Se supply in coffee plants cultivated in field conditions, Mateus et al. [27] found that Se can protect the photosynthetic pigments and increase coffee bean yield. Moreover, Luo et al. [28] showed that Se increased photosynthetic parameters during OS in rice. Also, the same authors found that Se can promote a higher transcript level of antioxidant-related genes. However, Se concentrations in soils vary widely in the earth’s crust. Selenium is an element that has several physiological and biochemical characteristics, such as the mitigation of different types of abiotic stress. 

Selenium content in plant tissue is driven mainly by the soil Se content and the chemical interactions that this element undergoes in soils [29]. Tropical soils are generally considered Se-poor environments, i.e., have ≤0.5 mg kg^−1^ Se [30], and the average Se concentration in soils worldwide is relatively low (~0.4 mg kg^−1^) [31]. Indeed, researchers have found Se deficiency in soils across various countries, including Brazil. Gabos et al. [32] found that Se content in soils from the São Paulo State in Brazil ranges from <0.08 to 1.61 mg kg^−1^, with a mean of 0.19 mg kg^−1^.

Studies determining the most effective time to apply Se for achieving OS mitigation have previously been poorly investigated in the literature. Yet, plant supplementation using Se before stress has been responsible for triggering metabolic responses in plants, inducing a priming effect [33]. Priming effects were first used to describe the application technique of nutrient and/or plant biostimulants in seeds to increase their vigor during germination [34]. However, applying biostimulants, such as Se, has been considered a resistance inducer strategy in plants and can be thought as a promising strategy for crop production in response to future climate changes [35,36,37].

In this paper, the foliar application of such biostimulant element is also called “priming” due to the preparation effect that it can promote in the plants and its implication on metabolic responses before the stress [38]. However, exogenous Se applied post-stress can also be used as a last resource to alleviate the side effects of the lack of water in plants, but the effects of Se on these conditions need to be clarified. Thus, this study aimed to investigate the role of the Se supply in improving OS tolerance in coffee seedlings while also assessing the best time for Se application. 

## 2. Results

### 2.1. Analysis of Se Content

Selenium content in leaves was significantly increased by foliar application. There was a statistical difference observed between all the treatments with Se application and the controls without Se supply (Figure 1). The Se content in the control treatments was 0.37 and 0.38 mg kg^−1^ DW for the treatments with stressed (+OS-Se) and non-stressed plants (−OS-Se), respectively. In contrast, the average Se content in the remaining treatments’ leaves was 1.95 mg kg^−1^ DW. The highest leaf Se content was found in the −4BOS treatment, i.e., 3.22 mg g^−1^ DW, which corresponded to eight times the content analyzed in the control treatments.

### 2.2. Antioxidant Enzymes (APX, CAT, GR, and SOD), H_2_O_2_, and MDA

There was no marked trend of OS on H_2_O_2_ and MDA, even when the control with stressed treatment was compared with the non-stressed one (Figure 2). In the H_2_O_2_ assays, even if OS is considered one of the main triggering agents of reactive oxygen species (ROS), there was no statistically significant difference observed between the treatments. On the other hand, the treatments with Se application −4BOS and +4AOS promoted higher values of MDA content, indicating that these treatments induced lipid peroxidation in the leaves. 

When the control treatments were compared, OS significantly reduced the activity of GR and SOD, but did not affect the activities of APX and CAT (Figure 2), i.e., there was no significant difference observed between the +OS-Se and −OS-Se treatments.

The Se application 8 days before the plants were submitted to OS (−8BOS) promoted higher APX, CAT, and SOD activity levels than the treatment +OS-Se. The Se application at −8BOS increased the activity of these enzymes in the order of 356.3%, 228.5%, 771.1%, and 266.5% compared with +OS-Se for APX, CAT, and SOD, respectively. A reduction in the GR enzyme was noticed in the treatments 0OS; +4AOS; +8AOS; and +OS-Se compared with the treatment −OS-Se. Plants that were pre-treated with Se (−8 and −4BOS) displayed a higher GR content compared to the plants that had only received OS and no Se application. Furthermore, the levels of GR activity detected in the −8 and −4BOS treatments were found to be equivalent to those found in the non-stressed plants. (Figure 2).

### 2.3. Carbohydrates, Protein, Amino Acids, and Proline 

Regardless of the Se supply and OS, the total free amino acids, reducing sugars, and sucrose content were unaffected (Figure 3). On the other hand, Se foliar supply increased, to some extent, the proline and protein content. The proline content obtained with the application of Se at −4BOS and +8AOS was significantly higher than that observed for the stressed plants without Se (+OS-Se). Hence, the Se supplementation could be seen as a strategy to increase these compounds in coffee leaves under OS.

On the other hand, the imposed OS affected the starch content, with all the treatments submitted to the stress showing lower starch content compared with the absolute control treatment (−OS-Se). However, all the treatments with Se application promoted lower starch content than the positive control treatment (+OS-Se), except for the treatment with the application on the day on which the stress was imposed (0OS) (Figure 3). Such results indicate that OS can reduce the starch content, but the Se supply can impose a lower starch content than that detected in plants subjected to OS without Se.

### 2.4. Chlorophyll Fluorescence Parameters (MultispeQ^®^)

The graph of chlorophyll was inserted as Appendix A. There was no statistically significant difference observed regarding the ECSt, Phi2, PhiNO, PhiNPQ, and qL. The supply of selenium on the same day the plants were submitted to the OS (0OS) and 8 days after the plants were submitted to the OS increased the LEF—linear electron flux—compared with the treatment without Se supply and OS (Appendix A). The Se application at +4AOS promoted the highest NPQt, showing that, in a certain way, Se can act to quench the excess of light energy.

### 2.5. Principal Component Analysis (PCA)

The variables reducing sugars, total free amino acids, sucrose, ECSt, Phi2, PhiNO, PhiNPQ, and qL were excluded in the PCA analysis as they all exhibited a low effect of the treatments, as shown in the univariate analysis. Furthermore, the addition of these variables to the PCA reduced the explanation of the variables to 43.1%. The contribution of the selected variables is shown in the Appendix A.

Results of the PCA are shown in Figure 4. The PCA explained 64.8% of the data variance, with the first axis (PC1) explaining 45.0%, and the second axis (PC2) 19.8%. The PC1 was affected mainly by APX, SOD, and starch, while the values of GR, CAT, MDA, proline, and protein were explained by the PC2 (Appendix A). The Se content in leaves showed a significant correlation with APX and SOD, but a low correlation with starch (Figure 4). This behavior was also noticed in the correlation matrix (Appendix A), in which Se and APX showed a positive and statistically significant correlation (R^2^ = 0.60, *p <* 0.05) and a negative and statistically significant correlation with starch (R^2^ = −0.69, *p <* 0.05) (Appendix A).

The biplot correlation clusters clearly distinguished the treatments and their respective correlations (Figure 4). The treatment −8BOS showed a clear correlation with GR and CAT, corroborating the previously shown results (Figure 2). The biplot correlation clusters also revealed a strong correlation of Se, APX, and SOD with the treatment related to the previous Se application 4 days before the stress (−4BOS). 

## 3. Discussion

Osmotic stress encompasses stress-induced decreasing water potential (Ψw) in plant cells [39]. Considering that the water flow moves towards the lowest Ψw, if the stress continues, the leaves start to lose water, reflecting in the Ψw in the leaf [40]. As a typical response to low water potential, the leaves of the coffee plants in this trial started to become wilted and flabby 5 days after the imposed stress, leading to leaf prostration due to the turgor loss during abiotic stress. Osmotic stress tolerance involves the maintenance of the plant’s water status and, hence, cell turgor. This condition may be achieved through stomatal regulation, decreasing transpiration loss or osmotic adjustment with the accumulation of osmoprotective substances, such as proline, glycine betaine, soluble proteins, and sugars, which help plants conserve their water status [41]. 

The results of Ψw (Appendix A) showed that all plants treated with PEG-6000 suffered from OS. The effect of OS is also illustrated in Figure 5. OS imposed mild stress in the treatments −8BOS, −4BOS, 0OS, and +OS-Se (Ψw from −1.5 MPa to −2.5 MPa). Meanwhile, the treatments +4AOS and +8AOS were subjected to severe stress (Ψw > −2.5 MPa) (Appendix A). According to Suma [42], non-susceptible plants can keep a minor reduction in Ψw (6.9%) compared with a higher reduction (14.4%) in susceptible genotypes of finger millet. Then, after the plants were submitted to the stress, the Se application may have acted as a stressor in coffee plants, leading those plants to higher water potential loss and potentializing the OS response.

Plants treated with Se at all times had a higher relative water content (RWC) than the controls, including at the turgor loss point (RWC_tlp_), whereas the turgor loss point (π_tlp_) was less negative in all the same plants. The π_tlp_ indicates the cell water potential inducing turgor pressure loss, which is crucial to maintain gas exchange and plant growth. Plants with a low π_tlp_ tend to maintain stomatal and hydraulic conductance, photosynthetic efficiency, and growth at a lower external water potential [43]. This parameter is thus correlated with the ability to tolerate stress rather than avoid it. 

Although it is considered that a more negative π_tlp_ improves drought tolerance, as described above, it is also suggested that a less negative π_tlp_ may be helpful, as it enables leaves to lose turgor quickly and close their stomata, and thereby maintain a high RWC_tlp_ [44]. This response pattern was observed in our study. Plants treated with Se showed a 20% higher RWC_tlp_ than untreated plants. According to DaMatta and Ramalho [7], coffee leaves usually have a high RWC_tlp_, regardless of water availability, to avoid stress rather than tolerate it. For the authors, this seemed to be more related to stomatal regulation and gas exchange maintenance than turgor. We suggest that in our experiment, Se helped the stomatal regulation in coffee plants under osmotic stress as a strategy to decrease transpiration rates. Similar results were related to yellow sweet clover under OS and Se addition [45].

A high RWC_tlp_ formed despite very low water potential is generally correlated with osmotic adjustment. However, our study did not observe an increase in the concentration of proline or soluble sugars as a standard response to stress or Se application (Figure 3). Furthermore, it has been reported for coffee leaves that the accumulation of proline and other solutes does not always correlate well with OS tolerance [7]. In our study, the application of Se 4 days before and 8 days after stress (−4BOS and +8AOS) seemed to have stimulated an osmotic adjustment due to the combination of a very low water potential, high relative water content, and proline accumulation concerning the controls (with and without stress). The high RWC_tlp_ in all treatments with Se application can be better explained by stomatal regulation, as mentioned before.

Stomatal closure in response to stress might limit CO_2_ absorption by the leaves. In our study, photosystem II efficiency showed no change in response to stress or Se (Appendix A). Associated with the fact that the plants did not show a reduction in growth, we can conclude that there was no photochemical limitation in photosynthesis. In line with this, we also observed no alteration in soluble sugars or sucrose in response to stress, suggesting no significant chemical limitations (Figure 3). Only starch was reduced in response to stress and Se application. 

In photosynthetic cells, starch is mostly synthesized using a fraction of the CO_2_-fixed carbon and temporarily stored in the chloroplast called “transitory”. The transitory starch is usually synthesized during the day and consumed at night to provide a constant flow of carbon and energy without photosynthesis [46]. Starch is considered the major carbohydrate storage in plants [47]. In stressful conditions, starch represents a pool of energy that can induce metabolic responses and help plants overcome harmful circumstances. It can be broken down into low molecular weight compounds. Starch degradation can be stimulated in response to osmotic stress to promote osmotic adjustment, which might explain the response to treatments in which OS was imposed. In addition to this, a noteworthy factor is that abscisic acid (ABA) biosynthesis is the primary signal for starch degradation in response to osmotic stress [48].

An improvement of carbohydrate metabolism and water status caused by Se application has been found by Rady et al. [22] in tomato plants. According to these authors, Se has been correlated with elevated activity levels of the antioxidative defense system components—both enzymatic and non-enzymatic—under an insufficient water supply. Furthermore, increased levels of osmoprotectants have been associated with a higher cellular relative water content and membrane stability index, resulting in reduced electrolyte leakage, lipid peroxidation, and oxidative stress biomarkers.

In the extensive literature survey conducted by Thalmann et al. [47], the authors discovered that in 23 of the 36 studies considered, leaf starch content was said to decrease in response to abiotic stress, regardless of the species assessed. This result highlights the importance of starch in providing energy to deal with abiotic stresses. Then, the starch catabolism displaces carbons to produce osmoprotectants that induce osmotic adjustments and stabilize proteins [49,50], and also promotes signals that induce stress responses [47]. Our findings are in line with the research conducted by Lee et al. [51], who also observed a notable reduction in starch content in white clover leaves when exposed to OS. This reduction in starch content has been believed to be part of the adaptation mechanism that enables rice plants to carry out basal metabolism, thereby countering the changes induced by OS in photosynthesis. 

The fact that Se application caused a more substantial reduction in starch content in coffee leaves under OS led us to the hypothesis that reduced starch accumulation during OS may be a plant strategy to maintain the flow of carbon and energy availability for growth during the harmful condition (Figure 3 and Figure 4) [52]. This assumption is supported by Malik et al. [53], who showed that the presence of Se stimulates a significant rise in α-amylase and β-amylase activity in mungbean, ultimately leading to the hydrolysis of starch. 

The higher Se content in the plants supplied with Se was expected, since Se supplementation in coffee plants via foliar application (and other plant species) has been studied in the literature [19,27]. Selenium can be supplied via seed, soil, and foliar application routes [54,55]. However, when applied at the same rate, foliar applications have been considered the most efficient way to increase Se content in plant tissue [54,56,57]. Since an active chemical chain builds the Se assimilation pathway, the addition of Se to stressed plants (+4AOS and +8AOS) possibly consumed the energy used to trigger metabolic responses that was supposed to be used to overcome the stress, making the plants unable to keep the Ψw at higher levels in the leaves.

Despite the beneficial effects of Se having been detailed in the literature, it can be toxic depending on the tissue levels and plant health condition [58,59]. Due to the chemical similarity of Se and S, selenate (SeO_4_^−^) is transported into the plants through sulfate transporters [21,60]. Since it is inside the plant cell, it is metabolized in the plastids via the sulfur assimilation pathway to selenocysteine (SeCys) or selenomethionine (SeMet) [61,62]. Se-SeO_4_^−^ is first assimilated by an active form via the enzyme adenosine triphosphate sulfurylase (APS) and APS-reductase (APR). Adenosine triphosphate sulfurylase binds selenate with adenosine triphosphate (ATP) to form adenosine 5′-phosphoselenate (APSe). After that, APSe is reduced to selenite by APR [20,21,63]. Selenite is then converted into SeCys and available to be converted into other organic compounds—like SeMet and proteins—or stocked in the vacuoles [64]. Notably, these Se-amino acids serve as precursors of ethylene, and the production of this phytohormone is enhanced under stress conditions collaborating with stomatal closure [45,65]. 

Excess organic Se, such as SeMet and SeCys, might cause toxicity to plant cells by forming malformed selenoproteins due to the replacement of Cys/Met with SeCys and SeMet in the peptide chain. Changing between Cys and SeCys changes the cellular protein’s structure by replacing the disulfide bond with a diselenide bond, which affects the peptide chains redox potentials [66]. Protein function might be compromised if the organic selenocompounds are non-specifically integrated into proteins in place of their sulfur (S) equivalents. This condition might trigger the plants’ negative responses and osmotic imbalances [67,68,69]. This result is also supported by the protein content in the leaves of the treatments +4AOS and +8AOS, in which the protein content was higher than in the stressed plants without Se supply (+OS-Se) (Figure 3). 

Several studies have shown the positive effect of Se on increasing antioxidant enzyme activities [27,70]. This wide antioxidant capacity is due to the promotion of the selenoproteins and the enzyme cofactor role. These compounds enhance the antioxidant enzymes, such as glutathione peroxidase (GPX) and glutathione reductase (GR), which combat ROS during plant osmotic imbalance under stressful situations. The positive correlation between Se and GPX has been described and implicated in the presence of Se-dependent GPX [71,72]. It may be an osmoprotective strategy to mitigate the harmful effects of abiotic stresses, such as drought [18,26], salinity [73], heavy metals [74], and low temperature [19]. 

Indeed, in this trial, GR was the enzyme that better responded to the application of Se, and only the treatment −8BOS was able to increase the content of APX, CAT, SOD, and GR at the same time. This result shows that prior Se supply is the best way to induce antioxidant activity to trigger metabolic responses to ROS while also stimulating priming responses against the upcoming oxidative stress. These results corroborated those of Silva et al. [75], who also found that Se foliar application can provide an enhanced antioxidant metabolism by increasing superoxide dismutase (SOD), catalase (CAT), ascorbate peroxidase (APX), and glutathione reductase (GR) activity. 

The major members of the ROS family include free radicals, like O•^−2^, OH•, and non-radicals, like H_2_O_2_ and O_2_, and they are continuously produced at basal levels under favorable conditions. Under this condition, their potential to cause harm is neutralized through various antioxidant mechanisms that scavenge them [76]. However, ROS can be produced in excess when plants suffer from long-term stress, promoting serious damage to the cells by inhibiting proteins, DNA synthesis, and other metabolic pathways [77]. 

In the ROS detoxification process, SOD is considered the first line of defense because it is responsible for converting the superoxide radical (O^2−^) into hydrogen peroxide (H_2_O_2_) and oxygen (O_2_) and thus reduces the risk of hydroxyl radical formation. As a second pathway to scavenge ROS in the cell, CAT catalyzes the dismutation of H_2_O_2_ into H_2_O and O_2_; meanwhile, APX and GR also help to scavenge the H_2_O_2_ into H_2_O using ascorbic acid (AA) and glutathione as a reducing agent [76,77]. 

The improvement of the enzymatic antioxidant system has been responsible for mitigating different abiotic stresses. For example, heavy metal exposure tends to induce the production of excessive ROS, which interact with macromolecules, such as DNA, proteins, and lipids, leading to a series of vicious processes together. These changes can alter cellular redox equilibrium and redox homeostasis [78]. A moderate exposure to lead (Pb) increased leaf SOD (251%), CAT (60%), and APX (537%) compared with the control [79]. The authors attributed this result to tentative plant metabolic changes to trigger key antioxidant enzyme responses to resist oxidative damage. 

In a vast literature review, Rajput et al. [80] pointed that the transgenic overexpression of different genes might improve the enzymatic activity in plants and increase their stress tolerance to adverse conditions. According to these authors, a specific gene from *Sedum alfredii* is responsible for increasing Cu/Zn-SOD activity, conferring Cd tolerance in *Arabidopsis* [81]. In another study, the gene SiCSD from *Saussurea involucrate* increased drought and cold tolerance in transgenic tobacco by promoting higher activities of SOD, CAT, and APX [82]. Overexpression of the ascorbate peroxidase gene (AgAPX1) from *Apium graveolens* enhanced ascorbate content, antioxidant capacity, and drought resistance in transgenic *Arabidopsis* [83]. 

Other non-enzymatic pathways have also been responsible for mitigating abiotic stresses by increasing the antioxidant system. Amino acids, proline, carbohydrates, and certain fungicide responses in plants have been attributed to stress alleviation in plants. In *Arabidopsis thaliana* leaves under drought stress, Sperdouli and Moustakas [84] reported the buildup and interaction of proline, anthocyanins, and soluble sugars retaining a strong antioxidant defense. In response to osmotic stress, soluble carbohydrates are synthesized, acting as osmoprotectants that stabilize cellular membranes and sustain turgor, avoiding overstress by ROS [85]. Proline operates as a non-enzymatic antioxidant by obtaining OH∙ through the H- on its amine group and is further decarboxylated [86]. 

Additionally, a recent study showed the effect of fungicides acting as a non-enzymatic antioxidant in lettuce [87]. The authors showed that the fungicide named fluazinam and its mixtures induced diversified changes in plant defense to increase ROS scavenging in lettuce. In this trial, the processes of fungicide degradation induced the activation of antioxidant enzymes (CAT, POD, and SOD), also inducing an antioxidant response in the plants. 

The effects of Se on the antioxidant system of plants under abiotic stresses have been extensively explored as the primary regulator of plant growth and yield under these conditions [59]. This condition was evident in our study, in which most of the found results can be explained by factors related to the antioxidant metabolism of the plant (Figure 4). What has also been well discussed is how, and to what extent, different doses of Se in other species, plant organs, and developmental stages affect plant metabolism.

## 4. Materials and Methods

### 4.1. Plant Materials and Study Site

The experiment was carried out in a greenhouse using arabica coffee seedlings (*Coffea arabica* cv. Catuaí), one of Brazil’s most traditional species. The cv. Catuaí is well known for its high beverage quality, good plant health, and high yield [88,89]. The plants used in the trial were at the age of 5–6 fully expanded leaves and were previously selected to keep uniformity and health. The plants were provided by the National Institute of Science and Technology of Coffee (INCT Café). 

The seedlings were produced in 1 L plant grow bags filled with subsoil + cattle manure at a ratio of 3:1, with 5 g of single superphosphate being added to each kilogram of the mixture. After the seedlings reached 5–6 fully expanded leaves, they were acclimated in a greenhouse at the Soil Science Department at the Federal University of Lavras (UFLA), located in Lavras, state of Minas Gerais, for 20 days. The greenhouse temperature was 25/15 °C day/night, and the relative humidity was 50/85% day/night. Irrigation was conducted daily with 80 mL of deionized water. 

After the acclimation, the substrate was removed from each root system, and the plants were transferred into 1 L black plastic pots with nutritive solution [90]. The nutritive solution was composed of the following: 2 mM NH_4_H_2_PO_4_, 6 mM KNO_3_, 4 mM Ca(NO_3_)_2_·4H_2_O, 2 mM MgSO_4_·7H_2_O, 50 µM H_3_BO_3_, 10 µM MnCl_2_·4H_2_O, 7.6 µM ZnSO_4_·7H_2_O, 8 µM CuSO_4_·5H_2_O, 0.40 µM Na_2_MoO_4_, 0.10 mM NaCl, 90 µM NaEDTA, and 89 µM FeSO_4_·7H_2_O, as described by Kane et al. [91]. All plants underwent an acclimatization process for two weeks by applying a 20% and 40% ionic strength, respectively, for each week. After that, the plants were randomly selected to compose the treatments. In accordance with Salgado et al. [92], the plants were kept at 40% of the ionic strength until the end of the trial. A polystyrene layer was used on top of the pots filled with nutritive solution to avoid algae growth into the nutrient solution. In addition, we used a system composed of an air compressor pump and clear PVC flexible tubing to keep the nutrient solution oxygenating during the experiment.

### 4.2. Experimental Design and Treatments

The experimental design was composed of a randomized block with seven treatments and four replicates. The treatments consisted of Se application through foliar supply on five different days compared with the day of induced osmotic stress to establish the best day to apply Se in coffee plants under induced OS. The treatments were: (i) eight days before induced osmotic stress (−8 BOS); (ii) four days before induced osmotic stress (−4 BOS); (iii) the same day of the induced osmotic stress (0 OS); (iv) four days after induced osmotic stress (+4 AOS); and (v) eight days after induced osmotic stress (+8 AOS). Two control treatments were also included: (vi) induced osmotic stress without Se (+OS-Se); and (vii) without stress and Se. Osmotic stress was induced using polyethylene glycol (PEG-6000). To induce the priming effect of Se foliar application and the alleviating effect, the treatments consisting of Se foliar application before the osmotic stress were called “priming treatments”, whereas plants treated after the osmotic stress were named “alleviated treatments”.

### 4.3. Application of Foliar Treatments

The foliar Se application was performed according to Sousa et al. [19]. In brief, the plants with Se application were sprayed with 5 mL of Se solution at 80 mg L^−1^ Se + 0.5% *v*/*v* of mineral oil, and the remaining plants were sprayed with a mineral oil solution at the rate of 0.5% *v*/*v*. On the day of application, the plants were moved to the outer part of the greenhouse to avoid contaminating the remaining plants. The Se source was Na_2_SeO_4_—Sigma Aldrich, 98.9%.

### 4.4. Osmotic Stress Imposition and Leaf Water Status

Polyethylene glycol with a molecular weight of 6000 (PEG-6000) was added to the nutritive solution according to Villela et al. [93] to induce the osmotic stress of −0.8 MPa. For this, 261.95 g L^−1^ of PEG-6000 was added into each plastic pot containing the nutritive solution in the respective treatment with stress. The osmotic potential was used based on previous tests with coffee considering the osmotic potential of −0.1; −0.2; −0.4; −0.6; −0.8; and −1.0 MPa. The osmotic potential of −0.8 MPa promoted a leaf water potential (Ψw) between −1.5–−2.5 MPa, considered moderate stress to the coffee plants [94]. The determination of the Ψw in each leaf was carried out with a Scholander pressure chamber (model 1000, PMS Instruments, Albany, NY, USA) [95] to confirm the leaf turgor on the day of sample collection and chlorophyll fluorescence parameter evaluation (Appendix A). The parameters elasticity, osmotic potential, relative water content, and turgor loss point were derived from PV curves, according to Tyree and Hammel [96]. 

### 4.5. Leaf Sample Collection and Preparation

All treatments’ leaf sample collection and photosynthetic parameters were performed seven days after the last Se foliar application (+8 AOS). The second fully expanded pair of leaves from top to bottom was used to perform the non-invasive analysis of the photosynthetic parameters (MultispeQ^®^) [97]. After the measurement, the leaves were collected and immediately snap frozen in liquid nitrogen, individually macerated in liquid nitrogen, homogenized in a cooled mortar using 0.1 g of the antioxidant polyvinylpyrrolidone (PVPP), and stored at −80 °C. The frozen samples were used to determine the analyses of lipid peroxidation (MDA content), hydrogen peroxide (H_2_O_2_), catalase (CAT), superoxide dismutase (SOD), glutathione reductase (GR), and ascorbate peroxidase (APX). 

The third and fourth fully expanded pairs of leaves from top to bottom were collected and washed three times with distilled water. All samples were dried at 65 °C for 72 h and were subjected to grinding in the Willey grinder. Ground samples were labeled and kept in air-tight plastic containers until they were used to quantify Se content, carbohydrates, protein, total free amino acids, and proline.

### 4.6. Determination of Examined Parameters

#### 4.6.1. Selenium Content in Leaves and Detection Limit (LOD and LOQ)

The Se content in the leaves was performed according to the USEPA 3051A protocol (U.S. Environmental Protection Agency—USEPA) with modifications [98]. Briefly, 0.5 g of dried leaf samples was digested with 5 mL of HNO_3_ in a microwave (Mars 5, CEM Corporation, Matthews, NC, USA). To avoid foaming and splashing, the vessels were kept in a cool room with a controlled temperature for 30 min after the end of the digestion program and opened carefully, and the volume was made up to 50 mL with water. A blank and a certified reference material for Se (white clover, BCR402-IRMM) were included in each batch of samples. The Se content in the leaves was measured using GFAAS (graphite furnace atomic absorption spectrometry), atomic absorption spectrometry with Zeeman background correction, and an EDL lamp for Se; Analyst™ 800 AAS, Perkin Elmer. The detection and quantification limits (LOD and LOQ) were determined according to Silva Junior et al. [99]. The LOD and LOQ for Se were 2.49 and 8.32 μg kg^−1^, respectively. The Se recovery rate in the reference material was 96.7% ± 1.28.

#### 4.6.2. Carbohydrates, Total Protein, and Total Free Amino Acids

The extraction of carbohydrates (starch, sucrose, and reducing sugars), total free amino acids, and proteins was based on Zanandrea et al. [100]. Dried samples were weighed (0.2 g) and mixed with 5 mL of 100 mM potassium phosphate buffer (pH 7.0) and then warmed in a water bath at 40 °C for 30 min. The solution was centrifuged at 10,000× *g* for 20 min and the supernatant was collected. This procedure was performed twice, and the supernatant was combined, totaling 10 mL. The first supernatant sample was used to quantify the carbohydrates and total free amino acids. The pellet was resuspended and used for starch extraction, mixing 8 mL of potassium acetate buffer (200 mM; pH 4.8) and 2 mL of amyloglucosidase (1 mg mL^−1^; 16 units of enzyme). 

The contents of starch and sucrose were determined using the anthrone method as follows: 30 µL of the supernatant was mixed with 2 mL of the ice-cold anthrone reagent (0.84 g of anthrone in 1 L of 63% sulfuric acid), and the mixture was heated in a boiling water bath for 3 min and cooled in ice. Absorbance was measured at 620 nm [101]. Reducing sugars were quantified using the 3,5-dinitrosalicylic acid (DNS) method as follows: 150 µL of the supernatant was mixed with 0.5 mL of DNS solution (2.50 g of DNS in 50 mL of NaOH 2N solution, 125 mL of distilled water, and 75 g of potassium sodium tartrate were heated using a water bath until completely dissolving and then diluted to 100 mL with distilled water) and 0.6 mL of distilled water. The mixture was heated in a boiling water bath for 5 min and water cooled. Absorbance was measured at 530 nm [102]. 

Total free amino acids were analyzed according to the ninhydrin method (0.2 mL of ninhydrin—5% *w*/*w*—in ethylene glycol monoethyl ether). For these measurements, 30 µL of the supernatant was mixed with 0.2 M citrate (pH 5.0), and 5% ninhydrin, 2% potassium cyanide, and 60% ethanol were added to the samples. Reactions were assessed using a spectrophotometer at 570 nm, and the results were compared with a standard curve of 0.1 μmol mL^−1^ glycine [103]. The protein content was determined using the Bradford assay as described by Bradford [104], with BSA applied as a protein standard. The analyses were carried out in duplicate and were measured using an Epoch^®^ Microplate Spectrophotometer (BioTek Instruments, Winooski, VT, USA).

#### 4.6.3. Proline

Proline content was estimated using the method described by Bates et al. [105]. Dried leaf samples (0.2 g) were weighed and macerated with 3% sulfosalicylic acid and heated in a water bath for 60 min at room temperature. After that, the samples were centrifuged at 10,000× *g* for 30 min. The supernatant (0.1 mL) was then mixed with 2 mL of acid ninhydrin (2.5g of ninhydrin in 40mL of phosphoric acid and 60mL of acetic acid) and determined using a colorimetric method (520 nm).

#### 4.6.4. Antioxidant Enzymes (SOD, CAT, APX, and GR)

The extraction of antioxidant enzymes was based on Biemelt et al. [106]. Frozen leaf samples were weighted (0.2 g) and mixed with 1.5 mL of potassium phosphate buffer solution (0.1 mol L^−1^, pH 7.8 + 0.1 mol L^−1^ EDTA, pH 7.0, 0.01 mol L^−1^ ascorbic acid, and 22 mg of PVPP). The solution was centrifuged at 13,000× *g* for 10 min at 4 °C. The enzymatic analyses’ quality assurance and quality control were warranted using two blanks in each reading plate and operating the samples at 0–4 °C. In addition, the enzyme extraction was performed on the day of the analysis to avoid the oxidation of the enzyme extract. The analyses were carried out in triplicate and were measured using an Epoch^®^ Microplate Spectrophotometer (BioTek Instruments, Winooski, VT, United States). The supernatant was used to quantify the activity of superoxide dismutase (SOD, EC: 1.15.1.1), catalase (CAT, EC: 1.11.1.6), ascorbate peroxidase (APX, EC: 1.11.1.11), and glutathione reductase (GR, EC: 1.8.1.7). 

The assay on SOD activity was performed by measuring its ability to inhibit the photochemical reduction of nitro blue tetrazolium at 560 nm [107]. Catalase (CAT) activity was assayed through measuring the rate of decomposition of H_2_O_2_ at 240 nm [108]. Ascorbate peroxidase (APX) was determined by reducing ascorbate at 290 nm [109]. Glutathione reductase (GR) was assayed according to the methodology proposed by Schaedle and Bassham [110] and adapted by García-Limones et al. [111].

#### 4.6.5. Hydrogen Peroxide and Lipid Peroxidation (Malondialdehyde)

The frozen leaf tissue (0.2 g) was homogenized in 5 mL of trichloroacetic acid (TCA), and centrifuged at 12,000× *g* for 15 min at 4 °C. The supernatant was collected to quantify the hydrogen peroxide according to Velikova [112] with modifications [113]. Lipid peroxidation analysis was assayed from the content of malondialdehyde (MDA) using thiobarbituric acid (TBAR) according to Buege and Aust [114] and Silva et al. [75].

#### 4.6.6. Chlorophyll Fluorescence Parameters (MultispeQ^®^)

The electron transport and electrochromic shift parameters were measured with the handheld unit MultispeQ^®^ using the PhotosynQ web application (https://photosynq.org; accessed on 15 April 2021) according to Kuhlgert et al. [97]. The following parameters were measured: total electrochromic shift (ECSt); linear electron flow (LEF); total non-photochemical quenching (NPQt); quantum yield of photosystem II II (Phi2); quantum yield of non-regulated energy loss in PSII (PhiNO), quantum yield of regulated non-photochemical quenching in PSII (PhiNPQ), and a fraction of PSII centers which are in the open state (qL).

### 4.7. Statistical Analysis and PCA

Generalized linear models (GLMs) were constructed to compare the treatments tested to each variable studied. The GLMs were used due to the non-uniformity of the residues of certain variables. After building the models, a Chi-squared test was performed to determine the differences that existed between treatments studied with the ANOVA function [115], complemented with multiple comparisons with the “ghlt function” [116]. The comparisons were carried out as follows: (I) Each treatment with water deficit was compared with the treatment without water deficit, and (II) comparison involved the different strategies of Se application compared with the cultivation with a water deficit. In addition, principal component analysis (PCA) was performed to determine the relationships among several variables. The variables were selected according to the main effects observed in the univariate analysis, and to increase the explained variance in the PCA. All statistical analyses were performed with the R software [117] using the base, stats, nlme, multcomp, FactoMineR, and factoextra packages [118,119,120].

## 5. Conclusions

OS, induced with PEG-6000, imposed significant stress on the *Coffea arabica* cv. Catuaí, promoting an imbalance in the water relationship. At the same time, OS reduced the GR and SOD activity compared with the control treatment. Selenium foliar supply revealed great potential for reducing the adverse effects of OS as a priming strategy 8 days before stress, improving the water relations, increasing the enzymatic activity (GR, SOD, and CAT), and potentiating the starch degradation under stress conditions. These findings also assist decision makers in how to deal with a foreseen drought in coffee plantations, where the earlier administration of foliar Se aids in the setting up of metabolic reactions that help the plants to combat the stress caused by a shortage of water. On the other hand, the post-stress application seems to impose extra stress on the plants, leading them to reduce their water potential. These results might support new nutritional strategies to induce stress responses in plants, leading to better plant development and sustainable crop production. To elucidate the role of Se on triggering metabolic responses in plants under OS, we suggest that other studies should be conducted to assess: (i) combined Se application with other nutrients; (ii) genetic assays; (iii) cross-species testing; and, (iv) long-term effects of Se in plants.

## Figures and Tables

**Figure 1 plants-12-03026-f001:**
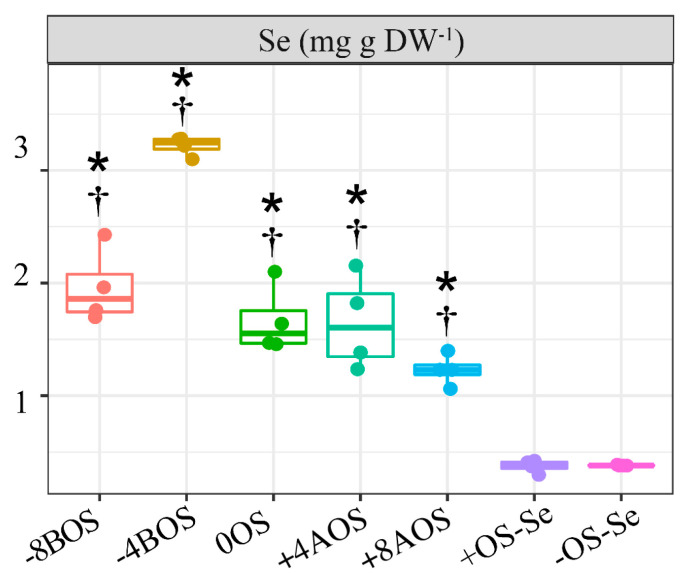
Leaf Se content as a result of Se application in *Coffea arabica* cv. Catuaí seedlings under osmotic stress induced with PEG-6000. The values displayed are the distribution of four replicates. Asterisks refer to the significant difference when comparing all treatments with non-stressed plants without Se supply (−OS-Se) (*p* < 0.05). Dagger refers to the significant difference when comparing all treatments with stressed plants without Se supply (+OS-Se) (*p* < 0.05). Treatments: −8BOS—application of Se 8 days before stress (stressed plants); −4BOS—application of Se 4 days before stress (stressed plants); 0OS—application of Se on the day of stress occurrence (stressed plants); +4AOS—application of Se 4 days after stress (stressed plants); +8AOS—application of Se 8 days after stress (stressed plants); +OS-Se—without Se (stressed plants); and −OS-Se—without Se (non-stressed plants).

**Figure 2 plants-12-03026-f002:**
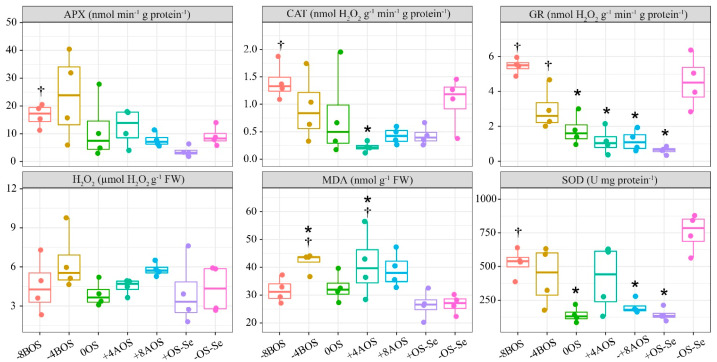
Hydrogen peroxide (H_2_O_2_), and lipid peroxidation (MDA) content, and activity of leaf antioxidant enzymes as a result of Se application in *Coffea arabica* cv. Catuaí seedlings under osmotic stress induced with PEG-6000. The values displayed are the distribution of four replicates. Asterisks refer to the significant difference when comparing all treatments with non-stressed plants without Se supply (−OS-Se) (*p* < 0.05). Dagger refers to the significant difference when comparing all treatments with stressed plants without Se supply (+OS-Se) (*p* < 0.05). Treatments: −8BOS—application of Se 8 days before stress (stressed plants); −4BOS—application of Se 4 days before stress (stressed plants); 0OS—application of Se on the day of stress occurrence (stressed plants); +4AOS—application of Se 4 days after stress (stressed plants); +8AOS—application of Se 8 days after stress (stressed plants); +OS-Se—without Se (stressed plants); and −OS-Se—without Se (non-stressed plants).

**Figure 3 plants-12-03026-f003:**
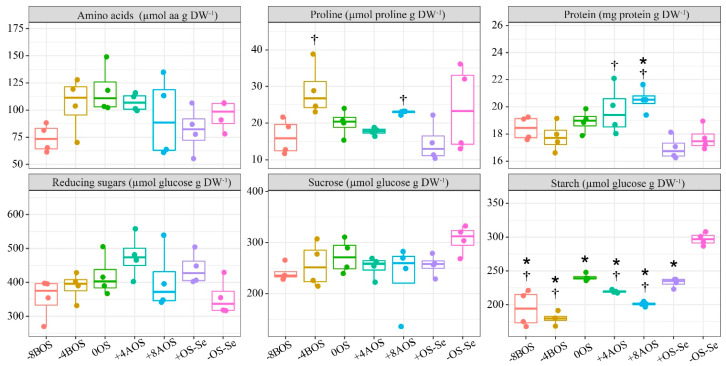
Total free amino acids (AA), proline (Pro), carbohydrates, and protein (Prt) as a result of Se application in *Coffea arabica* cv. Catuaí seedlings under osmotic stress induced with PEG-6000. The values displayed are the distribution of four replicates. Asterisks refer to the significant difference when comparing all treatments with non-stressed plants without Se supply (−OS-Se) (*p* < 0.05). Dagger refers to the significant difference when comparing all treatments with stressed plants without Se supply (+OS-Se) (*p* < 0.05). Treatments: −8BOS—application of Se 8 days before stress (stressed plants); −4BOS—application of Se 4 days before stress (stressed plants); 0OS—application of Se on the day of stress occurrence (stressed plants); +4AOS—application of Se 4 days after stress (stressed plants); +8AOS—application of Se 8 days after stress (stressed plants); +OS-Se—without Se (stressed plants); and −OS-Se—without Se (non-stressed plants).

**Figure 4 plants-12-03026-f004:**
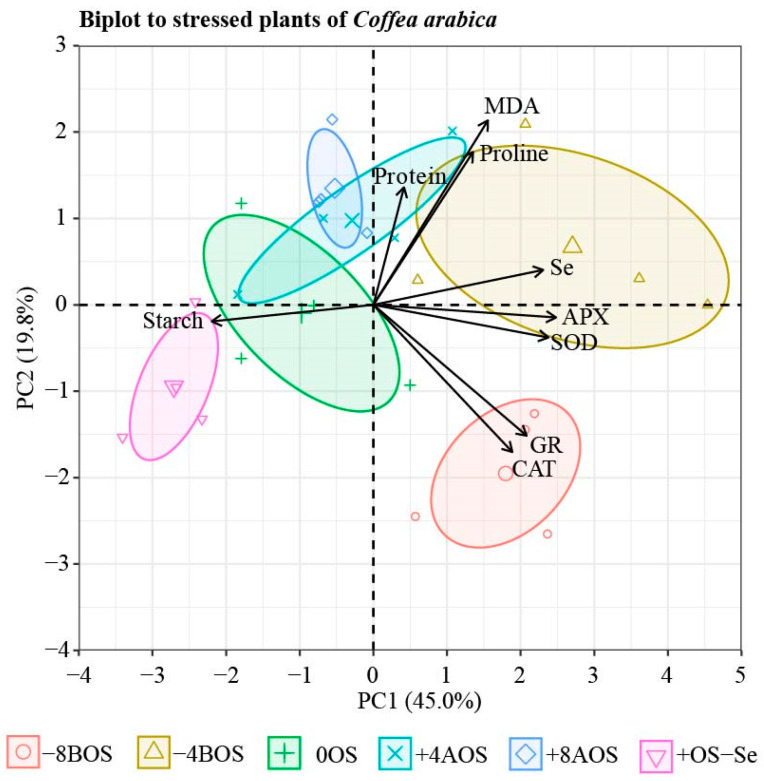
Principal component analysis (PCA) of leaf compounds and Se content in leaves. The leaf attributes included were leaf Se content (Se); ascorbate peroxidase (APX); superoxide dismutase (SOD), catalase (CAT), glutathione reductase (GR); proline; protein; lipid peroxidation (MDA), and starch. Arrows represent the contribution of leaf compounds on the principal component axes. Treatments: −8BOS—application of Se 8 days before stress (stressed plants); −4BOS—application of Se 4 days before stress (stressed plants); 0OS—application of Se on the day of stress occurrence (stressed plants); +4AOS—application of Se 4 days after stress (stressed plants); +8AOS—application of Se 8 days after stress (stressed plants); +OS-Se—without Se (stressed plants); and −OS-Se—without Se (non-stressed plants).

**Figure 5 plants-12-03026-f005:**
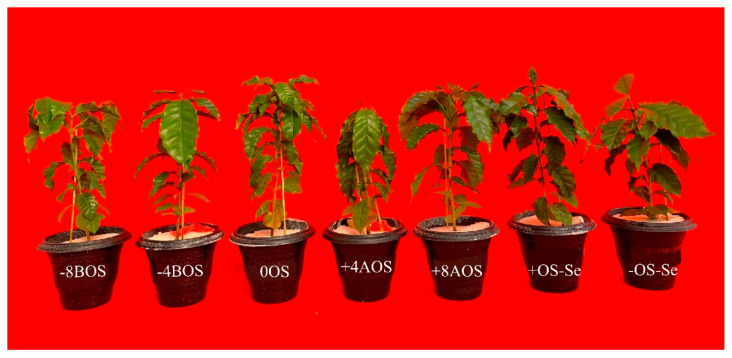
*Coffea arabica* cv. Catuaí seedlings under osmotic stress induced with PEG-6000 and Se foliar application. Treatments: −8BOS—application of Se 8 days before stress (stressed plants); −4BOS—application of Se 4 days before stress (stressed plants); 0OS—application of Se on the day of stress occurrence (stressed plants); +4AOS—application of Se 4 days after stress (stressed plants); +8AOS—application of Se 8 days after stress (stressed plants); +OS-Se—without Se (stressed plants); and −OS-Se—without Se (non-stressed plants).

## Data Availability

The data presented in this study are available from the corresponding author upon reasonable request.

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
