# Peer review of "Foliar Selenium Application to Reduce the Induced-Drought Stress Effects in Coffee Seedlings: Induced Priming or Alleviation Effect?"

_plants, 2023, doi:10.3390/plants12173026_

Round 1

Reviewer 1 Report

Overall comment:

The authors have identified an important and interesting issue. Scientifically the MS is strong and I recommend its publication in “plants-MDPI journal” after minor revision. English is not well presented. Even some sentences are difficult to understand.

-Authors should have a strong justification for choosing Selenium application only. Why not others? It should be mentioned more elaborately in the introduction part itself.

-Similarly, authors need to justify for selection of foliar application of selenium to coffee on physiological and biochemical aspects more deeply. 

Specific Comments:

Abstract: Needs modification. Please include some more numerical data in the abstract.

Keywords: I suggest authors add more keywords

Introduction:

As suggested, please put strong justification for choosing Se and Coffee in the introduction part.

Materials and Methods:

Please revise this section critically.

Only single Se concentration applied at different days? If yes why?

Line 351: ………….. high health means what?

Line 355: ………. they were installed……. Installed means what?

Line 360: ……….. black plastic pots….. please mention the capacity of filled soil?

Section 2.8- Please rewrite to focus about the statistics more deeply.

Results and Discussion: Well written but still needs more deep discussion with recent and relevant literatures.

Conclusion should also focus future scope of the study.

Language should be improved

Author Response

Review Report Forms – Reviewer I (The Reviewer I’s recommendations are in yellow)

OVERALL COMMENT:

The authors have identified an important and interesting issue. Scientifically the MS is strong and I recommend its publication in “plants-MDPI journal” after minor revision. English is not well presented. Even some sentences are difficult to understand.

Reply:  English was more elaborated, and some sentences were rewritten to clarify the intention.

-Authors should have a strong justification for choosing Selenium application only. Why not others? It should be mentioned more elaborately in the introduction part itself.

Reply: The introduction was improved adding the reason for applying only Se and not other nutrients (line 67- 71)

“Selenium can also foster synergistic interactions with other nutrients, all of which con-tribute to improved resilience against abiotic stresses and overall better plant growth [23,24]. However, the study of the effect of isolated Se applications remains to be clarified in specific conditions as a first step to define its effects on the main crops of interest.”

-Similarly, authors need to justify for selection of foliar application of selenium to coffee on physiological and biochemical aspects more deeply. 

Reply:  The effects of foliar Se application on physiological and biochemical aspects was more described in the introduction.

SPECIFIC COMMENTS:

Abstract: Needs modification. Please include some more numerical data in the abstract.

Reply: Numerical data were included in the abstract (lines 20-21,24).

Keywords: I suggest authors add more keywords

Reply: The keywords were modified, and “Oxidative Stress” and “Osmotic Potential” were added. (lines 29-30)

“Beneficial Elements; Oxidative Stress; Tropical Agriculture; Coffee Belt; Osmotic Potential.”

Introduction:

As suggested, please put strong justification for choosing Se and Coffee in the introduction part.

Reply: Some sentences were introduced throughout the topic to make it more meaningful and stronger.

Materials and Methods:

Please revise this section critically.

Only single Se concentration applied at different days? If yes why?

Reply: Yes. Only one single Se concentration was applied at different days. The concentration used in the trial was validated in previous studies and showed the most efficient rate to help plants against abiotic stresses. The use of many concentrations could stifle the results of the Se application in different days.

- Line 351: ………….. high health means what?

Reply: The expression “high health” was shifted to the word “health” to improve the understanding (line 397).

Line 355: ………. they were installed……. Installed means what?

Reply: The sentence was rewritten to improve the understanding (lines 401 – 403)

“…they were acclimated in a greenhouse at the Soil Science Department at the Federal Uni-versity of Lavras (UFLA), located in Lavras, State of Minas Gerais, for 20 days.”

Line 360: ……….. black plastic pots….. please mention the capacity of filled soil?

Reply: The sentence was rewritten (line 407).

“…1-L black plastic pots with nutritive solution.”

Section 2.8- Please rewrite to focus about the statistics more deeply.

Reply: The section “Principal component analysis (PCA)” was rewritten to improve the understanding (lines 196 – 211).

Results and Discussion: Well written but still needs more deep discussion with recent and relevant literatures.

Reply: Some sentences were included in the topic to add more discussion and relevant literatures (lines 196 – 211; 299 – 305; 311 – 316; 320 – 323)

Conclusion should also focus future scope of the study.
Reply:
Some sentences were added to the conclusion in order to focus on the futures scope of the study (lines 587 – 592).

“These results might support new nutritional strategies to induce stress responses in plants, leading to better plant development and sustainable crop production. To elucidate the role of Se on triggering metabolic responses in plants under OS, the authors suggest that other studies should be conducted to access: i) Combined Se application with other nutrients; ii) Genetic assays; iii) Cross-species testing; and, iv) Long-term effects of Se in plants.”

Comments on the Quality of English Language: Language should be improved

Reply: The language was revised to improve quality of the English.  

Reviewer 2 Report

The manuscript deals with the evaluation of different time of selenium application in coffee plants response to drought stress. The paper emphasizes the importance of time in the application of compounds which can alleviate abiotic stress. Despite an important assumption of the study, the manuscript has some flaws. In the description of the results, values or % differences between treatments are needed. The details are listed below:

L30-33: add some % changes of examined parameters between treatments

L70: indicate the plant response to drought stress at the level of antioxidant enzymes. Emphasize the ubiquitous role of antioxidant system in the mitigation of different abiotic stresses (drought, temperature, salinity, pesticides, heavy metals). For this purpose the Authors may refer to the following reference: https://doi.org/10.1016/j.scienta.2022.110988

L90-95: expand description of the results emphasizing the highest Se content in -4BOS

L96: in the Fig. 1 correct the names of x axis. +4BOS and +8BOS replace with +4AOS and +8AOS

L103: ‘0OS - Application of Se in stress’ – replace with ‘Application of Se on the day of stress occurrence’ (or something similar). Correct also in Fig. 2-5

L107: rearrange the description of these parameters with the order of parameters in Fig. 2

L112: ‘higher values of its content’ - ‘higher values of MDA content’

L123-124: clarify to express the meaning of Fig. 2

L184-186: indicate correlation coefficients. The Authors should consider to include the Table or heatmap Figure with correlation coefficients which would improve the value of obtained results

L276: ABA – full name

L330-331: rephrase

L360: describe the composition of nutritive solution

L421: replace ‘Assessments’ with ‘Determination of examined parameters’

L446-449: describe briefly

L455: indicate the concentration and volume of ninhydrin

L495: ‘residues of some variables’ – rephrase

Author Response

Review Report Forms – Reviewer II (The Reviewer II’s recommendations are in blue)

Comments and Suggestions for Authors

The manuscript deals with the evaluation of different time of selenium application in coffee plants response to drought stress. The paper emphasizes the importance of time in the application of compounds which can alleviate abiotic stress. Despite an important assumption of the study, the manuscript has some flaws. In the description of the results, values or % differences between treatments are needed. The details are listed below:

L30-33: add some % changes of examined parameters between treatments

Reply:  Some % changes were inserted in the abstract as part of the Reviewer I’s suggestions (in yellow) (line 20-21; 24)

L70: indicate the plant response to drought stress at the level of antioxidant enzymes. Emphasize the ubiquitous role of antioxidant system in the mitigation of different abiotic stresses (drought, temperature, salinity, pesticides, heavy metals). For this purpose the Authors may refer to the following reference: https://doi.org/10.1016/j.scienta.2022.110988

Reply: The topic was improved, and it was very helpful. The suggestions were made, and the citation was included in the manuscript.

L90-95: expand description of the results emphasizing the highest Se content in -4BOS

Reply: It was explained in the manuscript (line 113 – 117)

The Se content in the control treatments was 0.37 and 0.38 mg kg-1 DW for the treatments with stressed (+OS-Se) and non-stressed plants (-OS-Se), In contrast, the average Se content in the remaining treatments' leaves was 1.95 mg kg-1 DW. The highest leaf Se content was found in the -4BOS treatment, i.e.,3.22 mg g-1 DW, which corresponded to 8 times the content analyzed in the control treatments.”

L96: in the Fig. 1 correct the names of x axis. +4BOS and +8BOS replace with +4AOS and +8AOS

Reply: The names of x axis in the Fig. 1 was corrected.

L103: ‘0OS - Application of Se in stress’ – replace with ‘Application of Se on the day of stress occurrence’ (or something similar). Correct also in Fig. 2-5

Reply: It was replaced in all figures as suggested (lines 125; 155 -156, 181-182; 219; 248 – 249).

“…Application of Se on the day of stress occurrence…”

L107: rearrange the description of these parameters with the order of parameters in Fig. 2

Reply: It was rearranged (line 129)

“2.2. Antioxidant enzymes (APX, CAT, GR, and SOD), H2O2, and MDA”

L112: ‘higher values of its content’ - ‘higher values of MDA content’

Reply:  It was replaced as suggested (lines 134 - 135)

L123-124: clarify to express the meaning of Fig. 2

Reply:  It was clarified (lines 144 - 147).

“Plants that were pre-treated with Se (-8 and -4BOS) showed higher GR content than plants that had only received OS and no Se application. Furthermore, the levels of GR activity detected in the -8 and -4BOS treatments were equivalent to those found in non-stressed plants. (Figure 2).”

L184-186: indicate correlation coefficients. The Authors should consider to include the Table or heatmap Figure with correlation coefficients which would improve the value of obtained results.

Reply: A correlation matrix was inserted in the supplementary material to support the data. The correlation coefficients were included in the text to improve the value of the results (lines             206 and 209).  

“This behavior was also noticed in the correlation matrix (supplementary table 4), in which Se and APX showed a positive and statistically significant correlation (R2=0.60, p < 0.05) and a negative and statistically significant correlation with starch (R2=-0.69, p < 0.05) (supplementary figure 4).”

L276: ABA – full name

Reply: “ABA” was replaced by its full name (line 297).

L330-331: rephrase

Reply: It was rephrased (line 365 - 372)

“Indeed, in this trial, GR was the enzyme that better responded to the application of Se, and only the treatment -8BOS was able to increase the content of APX, CAT, SOD, and GR at the same time. This result shows that prior Se supply is the best way to induce antioxidant activity to trigger metabolic responses to ROS, while also stimulating priming responses against the upcoming oxidative stress. These results corroborated those of Silva et al., [75], who also found that Se foliar application can provide an enhanced antioxidant metabolism by increasing superoxide dismutase (SOD), catalase (CAT), ascorbate peroxidase (APX) and glutathione reductase (GR) activity.”

L360: describe the composition of nutritive solution

Reply:  It was indicated as suggested (lines 407 - 411)

“The nutritive solution was composed by 2 mM NH4H2PO4, 6 mM KNO3, 4 mM Ca(NO3)2·4H2O, 2 mM MgSO4·7H2O, 50 µM H3BO3, 10 µM MnCl2·4H2O, 7.6 µM ZnSO4·7H2O, 8 µM CuSO4·5H2O, 0.40 µM Na2MoO4, 0.10 mM NaCl, 90 µM NaEDTA, and 89 µM FeSO4·7H2O as described by Kane et al [81]

L421: replace ‘Assessments’ with ‘Determination of examined parameters’

Reply: It was replaced as suggested (line 471).

L446-449: describe briefly

Reply: The methodology was described more deeply (lines 498 – 516).

L455: indicate the concentration and volume of ninhydrin

Reply:  It was indicated as suggested (line 510 – 511; 522 - 524)

L495: ‘residues of some variables’ – rephrase

Reply: It was rephrased (line 563 - 565)

Reviewer 3 Report

In the manuscript "Foliar selenium application to reduce the induced-drought stress effects in coffee seedlings: induced priming or alleviation effect?", the authors tested the post-stress exogenous application of Se as a resource of last resort to mitigate the side effects of drought in coffee seedlings and evaluated the best application time.

This study is an interesting and clear with a valid and better selection of samples. The study included well-presented data and analysis, and figures are clarified. However, minor revisions are needed as follows:

- The abstracts section is long (306 words), please shorten it to the standard word count as in the 'Plants' guidelines.

- Line 19: ……. content of plant osmoprotectants, …..

- In the introduction, the authors wrote using many tenses, for example, in line 54 "The OS promoted changes in ……" and in line 55 " Water deficit directly affects crops' growth, ……". Please be consistent.

- Lines 62-65: please cite the following article:

"Rady, M.M.; Belal, H.E.E.; Gadallah, F.M.; Semida, W.M. Selenium application in two methods elevates drought tolerance in Solanum lycopersicum by increasing yield, quality, and antioxidant defense system and suppressing oxidative stress biomarkers. Sci. Hortic. 2020, 266, 109290."

- Lines 61-70: The improving effects of Se on stressed plants need more in-depth details.

- Line 93 and 95: mg kg DW-1 should be changed to mg kg-1 DW.

- Line 137: ……, Amino Acids, and Proline

- Line 258: "But we believe that …… ", the use of “we” or any other pronoun is not scientific, please change. And also if it finds anywhere else.

- Lines 274 and 275: "osmotic stress" and "OS", please be consistent.

- Lines 357-358: what about the precipitation in the study area?

- Line 376: (+4 AOS); and v) eight days after ……….

Minor editing of The English is needed.

Author Response

Review Report Forms – Reviewer III (The Reviewer III’s recommendations are in green)

Comments and Suggestions for Authors

In the manuscript "Foliar selenium application to reduce the induced-drought stress effects in coffee seedlings: induced priming or alleviation effect?", the authors tested the post-stress exogenous application of Se as a resource of last resort to mitigate the side effects of drought in coffee seedlings and evaluated the best application time.

This study is an interesting and clear with a valid and better selection of samples. The study included well-presented data and analysis, and figures are clarified. However, minor revisions are needed as follows:

- The abstracts section is long (306 words), please shorten it to the standard word count as in the 'Plants' guidelines.

Reply: The abstract was rewritten to fits on the Plants’ guidelines (lines 15-27).

- Line 19: ……. content of plant osmoprotectants, …..

Reply: This term was excluded from the abstract.

- In the introduction, the authors wrote using many tenses, for example, in line 54 "The OS promoted changes in ……" and in line 55 " Water deficit directly affects crops' growth, ……". Please be consistent.

Reply: The term “drought stress” was avoided in the text.

- Lines 62-65: please cite the following article:

"Rady, M.M.; Belal, H.E.E.; Gadallah, F.M.; Semida, W.M. Selenium application in two methods elevates drought tolerance in Solanum lycopersicum by increasing yield, quality, and antioxidant defense system and suppressing oxidative stress biomarkers. Sci. Hortic. 2020, 266, 109290."

Reply: The authors considered the paper’s results as very interesting and useful. The reference was used to improve not only the “introduction topic” but also the “discussion” of the results obtained (lines 63 – 67; 299 - 305)

“In a recent study, Rady et al. [22] found that Se was responsible for mitigating adverse effects of water-deficit stress conditions in Solanum lycopersicum. According to the authors, the improvements observed in response to Se indicate that it plays a regulatory role in plants under stress by positively influencing both enzymatic and non-enzymatic components of the plant's antioxidative defense system.”

“An improvement of carbohydrate metabolism and water status caused by Se application has been found by Rady et al. [22] in tomato plants. According to the authors, Se has been correlated with elevated activity levels of antioxidative defense system components - both enzymatic and non-enzymatic - under insufficient water supply. Further-more, increased levels of osmoprotectants have been associated with higher cellular relative water content and membrane stability index, resulting in reduced electrolyte leakage, lipid peroxidation, and oxidative stress biomarkers.”

- Lines 61-70: The improving effects of Se on stressed plants need more in-depth details.

Reply: More details about Se and its effects in plants were inserted in the topic (lines 57 – 91).

- Line 93 and 95: mg kg DW-1 should be changed to mg kg-1 DW.

Reply: It was corrected (line 113 – 117).

- Line 137: ……, Amino Acids, and Proline

Reply: The word “and” was added to the topic (line 160).

- Line 258: "But we believe that …… ", the use of “we” or any other pronoun is not scientific, please change. And also if it finds anywhere else.

Reply: The sentence was rewritten to exclude the word personal point of view (lines 279 – 280).

- Lines 274 and 275: "osmotic stress" and "OS", please be consistent.

Reply: It was corrected.

- Lines 357-358: what about the precipitation in the study area?

Reply: The sentence “The irrigation was made dairy with 80 mL of deionized water.” was added to the topic (lines 404 – 405).

 - Line 376: (+4 AOS); and v) eight days after ……….

Reply: the word “and” was added to the sentence (line 427).

Round 2

Reviewer 1 Report

The authors incorporated all corrections nicely. No further corrections are required. The article can be acceptable in its present form.

Author Response

According to the Reviewer I’s comments and suggestions, no further corrections are required for the Round II. 

Reviewer 2 Report

The Authors have included most of the comments in the revised manuscript. However, for example in L84 or L376 the Authors should add a general information related to the ubiquitous role of antioxidant system in the mitigating different abiotic stresses including pesticides or heavy metals. For this purpose the Authors may refer to the following reference: https://doi.org/10.1016/j.scienta.2022.110988

Author Response

Comments and Suggestions for Authors - (The Reviewer II’s recommendations are in purple)

- The Authors have included most of the comments in the revised manuscript. However, for example in L84 or L376 the Authors should add a general information related to the ubiquitous role of antioxidant system in the mitigating different abiotic stresses including pesticides or heavy metals. For this purpose the Authors may refer to the following reference: https://doi.org/10.1016/j.scienta.2022.110988.

Reply:
The suggestion was included in the topic (lines 388-422). 

Round 3

Reviewer 2 Report

The Authors have improved the paper. I have no more comments.